# EFFICIENT GAN-BASED ANOMALY DETECTION

**Houssam Zenati**[1,2]**, Chuan-Sheng Foo**[2]**, Bruno Lecouat**[2,3]
**Gaurav Manek**[4]**, Vijay Ramaseshan Chandrasekhar**[2,5]

[1] CentraleSupélec, `houssam.zenati@student.ecp.fr`.

[2] Institute for Infocomm Research, Singapore, {`foocs,vijay`}`@i2r.a-star.edu.sg`.

[3] Télécom ParisTech, `bruno.lecouat@gmail.com`.

[4] Carnegie Mellon University, `gaurav_manek@cmu.edu`.

[5] School of Computer Science, Nanyang Technological University.

## ABSTRACT

Generative adversarial networks (GANs) are able to model the complex high-dimensional distributions of real-world data, which suggests they could be effective for anomaly detection. However, few works have explored the use of GANs for the anomaly detection task. We leverage recently developed GAN models for anomaly detection, and achieve state-of-the-art performance on image and network intrusion datasets, while being several hundred-fold faster at test time than the only published GAN-based method.

## 1 INTRODUCTION

Anomaly detection is one of the most important problems across a range of domains, including manufacturing (Mart et al., 2015), medical imaging and cyber-security (Schubert et al., 2014). Fundamentally, anomaly detection methods need to model the distribution of normal data, which can be complex and high-dimensional. Generative adversarial networks (GANs) (Goodfellow et al., 2014) are one class of models that have been successfully used to model such complex and high-dimensional distributions, particularly over natural images (Radford et al., 2016).

Intuitively, a GAN that has been well-trained to fit the distribution of normal samples should be able to reconstruct such a normal sample from a certain latent representation and also discriminate the sample as coming from the true data distribution. However, as GANs only implicitly model the data distribution, using them for anomaly detection necessitates a costly optimization procedure to recover the latent representation of a given input example, making this an impractical approach for large datasets or real-time applications.

In this work, we leverage recently developed GAN methods that simultaneously learn an encoder during training (Vincent Dumoulin & Courville, 2017; Donahue et al., 2017) to develop an anomaly detection method that is efficient at test time.We apply our method to an image dataset (MNIST) (LeCun et al., 1998) and a network intrusion dataset (KDD99 10percent) (Lichman, 2013) and show that it is highly competitive with other approaches. To the best of our knowledge, our method is the first GAN-based approach for anomaly detection which achieves state-of-the-art results on the KDD99 dataset. An implementation of our methods and experiments is provided at `https://github.com/houssamzenati/Efficient-GAN-Anomaly-Detection.git`.

## 2 RELATED WORK

Anomaly detection has been extensively studied, as surveyed in (Chandola et al., 2009). Popular techniques utilize clustering approaches or nearest neighbor methods (Xiong et al., 2011; Zimek et al., 2012) and one class classification approaches that learn a discriminative boundary around normal data, such as one-class SVMs (Yunqiang Chen & Huang, 2001). Another class of methods uses fidelity of reconstruction to determine whether an example is anomalous, and includes Principal Component Analysis (PCA) and its kernel and robust variants (Jolliffe, 1986; S. Gnter & Vishwanathan, 2007; Candès et al., 2009). More recent works use deep neural networks, which do not require explicit feature construction unlike the previously mentioned methods. Autoencoders,

variational autoencoders (An & Cho, 2015; Zhou & Paffenroth, 2017), energy based models (Zhai et al., 2016) and deep autoencoding Gaussian mixture models (Bo Zong, 2018) have been explored for anomaly detection. Aside from AnoGAN (Schlegl et al., 2017), however, the use of GANs for anomaly detection has been relatively unexplored, even though GANs are suited to model the high-dimensional complex distributions of real-world data (Creswell et al., 2017).

## 3 EFFICIENT ANOMALY DETECTION WITH GANS

Our models are based on recently developed GAN methods (Donahue et al., 2017; Vincent Dumoulin & Courville, 2017) (specifically BiGAN), and simultaneously learn an encoder $E$ that maps input samples $x$ to a latent representation $z$, along with a generator $G$ and discriminator $D$ during training; this enables us to avoid the computationally expensive step of recovering a latent representation at test time. Unlike in a regular GAN where the discriminator only considers inputs (real or generated), the discriminator $D$ in this context also considers the latent representation (either a generator input or from the encoder).

Vincent Dumoulin & Courville (2017) explored different training strategies to learn an encoder such that $E = G^{-1}$, and emphasized the importance of learning $E$ jointly with $G$. We therefore adopted a similar strategy, solving the following optimization problem during training: $\min_{G,E} \max_D V(D, E, G)$, with $V(D, E, G)$ defined as

$$V(D, E, G) = \mathbb{E}_{x \sim p_X} \left[ \mathbb{E}_{z \sim p_E(\cdot|x)} \left[ \log D(x, z) \right] \right] + \mathbb{E}_{z \sim p_Z} \left[ \mathbb{E}_{x \sim p_G(\cdot|z)} \left[ 1 - \log D(x, z) \right] \right].$$

Here, $p_X(x)$ is the distribution over the data, $p_Z(z)$ the distribution over the latent representation, and $p_E(z|x)$ and $p_G(x|z)$ the distributions induced by the encoder and generator respectively.

Having trained a model on the normal data to yield $G, D$ and $E$, we then define a score function $A(x)$ (as in Schlegl et al. (2017)) that measures how anomalous an example $x$ is, based on a convex combination of a reconstruction loss $L_G$ and a discriminator-based loss $L_D$:

$$A(x) = \alpha L_G(x) + (1 - \alpha) L_D(x)$$

where $L_G(x) = ||x - G(E(x))||_1$ and $L_D(x)$ can be defined in two ways. First, using the cross-entropy loss $\sigma$ from the discriminator of $x$ being a real example (class 1): $L_D(x) = \sigma(D(x, E(x)), 1)$, which captures the discriminator's confidence that a sample is derived from the real data distribution. A second way of defining the $L_D$ is with a "feature-matching loss" $L_D(x) = ||f_D(x, E(x)) - f_D(G(E(x)), E(x))||_1$, with $f_D$ returning the layer preceding the logits for the given inputs in the discriminator. This evaluates if the reconstructed data has similar features in the discriminator as the true sample. Samples with larger values of $A(x)$ are deemed more likely to be anomalous.

## 4 EXPERIMENTS

We provide details of the experimental setup and network architectures used in the Appendix.

**MNIST:** We generated 10 different datasets from MNIST by successively making each digit class an anomaly and treating the remaining 9 digits as normal examples. The training set consists of 80% of the normal data and the test set consists of the remaining 20% of normal data and all of the anomalous data. All models were trained only with normal data and tested with both normal and anomalous data. As the dataset is imbalanced, we compared models using the area under the precision-recall curve (AUPRC). We evaluated our method against AnoGAN and the variational auto-encoder (VAE) of (An & Cho, 2015). We see that our model significantly outperforms the VAE baseline. Likewise, our model outperforms AnoGAN (Figure 1), but with approximately 800x faster inference time (Table 2). We also observed that the feature-matching variant of $L_D$ used in the anomaly score performs better than the cross-entropy variant, which was also reported in Schlegl et al. (2017), suggesting that the features extracted by the discriminator are informative for anomaly detection.

**KDD99:** We evaluated our method on this network activity dataset to show that GANs can also perform well on high-dimensional, non-image data. We follow the experimental setup of (Zhai et al., 2016; Bo Zong, 2018) on the KDDCUP99 10 percent dataset (Lichman, 2013). Due to the

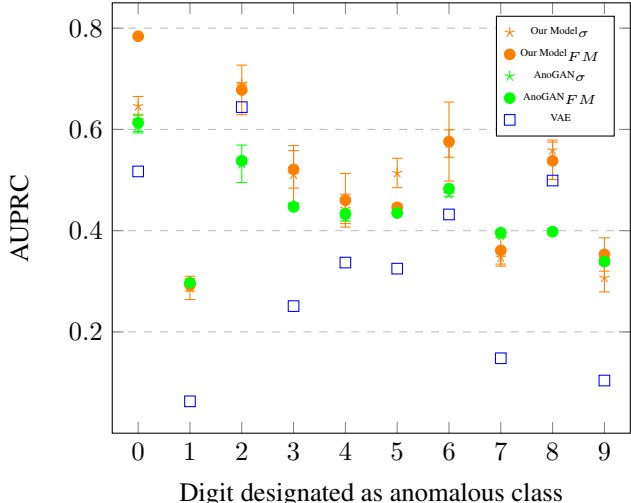

Figure 1: Performance on MNIST measured by the area under the precision-recall curve. Best viewed in color. VAE data were obtained from (An & Cho, 2015). Error bars show variation across 3 random seeds. FM and $\sigma$ respectively denote feature-matching and cross-entropy variants of $L_D$ used in the anomaly score.

proportion of outliers in the dataset "normal" data are treated as anomalies in this task. The 20% of samples with the highest anomaly scores $A(x)$ are classified as anomalies (positive class), and we evaluated precision, recall, and F1-score accordingly. For training, we randomly sampled 50% of the whole dataset and the remaining 50% of the dataset was used for testing. Then, only data samples from the normal class were used for training models, therefore, all anomalous samples were removed from the split training set. Our method is overall highly competitive with other state-of-the art methods and achieves higher recall. Again, our model outperforms AnoGAN and also has a 700x to 900x faster inference time (Table 2).

Table 1: Performance on the KDD99 dataset. Values for OC-SVM, DSEBM, DAGMM values were obtained from (Zhai et al., 2016; Bo Zong, 2018). Values for AnoGAN and our model are derived from 10 runs.

| **Model** | Precision | Recall | F1 |
|---|---|---|---|
| OC-SVM | 0.7457 | 0.8523 | 0.7954 |
| DSEBM-r | 0.8521 | 0.6472 | 0.7328 |
| DSEBM-e | 0.8619 | 0.6446 | 0.7399 |
| DAGMM-NVI | 0.9290 | 0.9447 | 0.9368 |
| DAGMM | **0.9297** | 0.9442 | 0.9369 |
| AnoGAN$_{FM}$ | $0.8786 \pm 0.0340$ | $0.8297 \pm 0.0345$ | $0.8865 \pm 0.0343$ |
| AnoGAN$_{\sigma}$ | $0.7790 \pm 0.1247$ | $0.7914 \pm 0.1194$ | $0.7852 \pm 0.1181$ |
| Our Model$_{FM}$ | $0.8698 \pm 0.1133$ | $0.9523 \pm 0.0224$ | $0.9058 \pm 0.0688$ |
| Our Model$_{\sigma}$ | $0.9200 \pm 0.0740$ | $\mathbf{0.9582 \pm 0.0104}$ | $\mathbf{0.9372 \pm 0.0440}$ |

## 5 CONCLUSION

We demonstrated that recent GAN models can be used to achieve state-of-the-art performance for anomaly detection on high-dimensional, complex datasets whilst being efficient at test time; our use of a GAN that simultaneously learns an encoder eliminates the need for a costly procedure to recover the latent representation for a given input. In future work, we plan to perform a more extensive evaluation of our method, evaluate other training strategies, as well as to explore the effects of encoder accuracy on anomaly detection performance.

ACKNOWLEDGMENTS

The authors would like to thank the Agency for Science, Technology and Research (A*STAR), Singapore for supporting this research with scholarships to Houssam Zenati and Bruno Lecouat. The authors also would like to thank Yasin Yazici for the fruitful discussions.

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

## APPENDIX

## A    EXPERIMENT DETAILS

We implemented AnoGAN and our BiGAN-based method in Tensorflow, and used the same hyperparameters as the original AnoGAN paper, using $\alpha = 0.9$ in the anomaly score $A(x)$ (both our model and AnoGAN) and running SGD for 500 iterations for AnoGAN. We attempted to match the AnoGAN and BiGAN architectures and learning hyperparameters as far as possible to enable a fair comparison. An exponential moving average of model parameters was used at test time, with a decay of 0.999 for the MNIST dataset and a decay of 0.9999 for the KDD99 dataset.

## B    INFERENCE TIME DETAILS

MNIST experiments were run on NVIDIA GeForce TitanX GPUs with Tensorflow 1.1.0 (python 3.4.3). KDD experiments were run on NVIDIA Tesla K40 GPUs with Tensorflow 1.1.0 (python 3.5.3).

Table 2: Average inference time over 100 batches (ms)

| Dataset | $L_D$ | AnoGAN | Our Model | Speed Up |
|---------|-------|--------|-----------|----------|
| MNIST   | $\sigma$ | 6657 | 8   | $\sim$830 |
|         | $FM$  | 7260   | 9.4 | $\sim$ 770 |
| KDD     | $\sigma$ | 2578 | 2.7 | $\sim$950 |
|         | $FM$  | 3527   | 5,3 | $\sim$660 |

## C  MNIST EXPERIMENTS DETAILS

**Preprocessing:** Pixels were scaled to be in range [-1,1].
The outputs of the starred layer in the discriminator were used for the *FM* scoring variant.

| Operation | Kernel | Strides | Features Maps/Units | BN? | Non Linearity |
|---|---|---|---|---|---|
| *G(z)* | | | | | |
| Dense | | | 1024 | √ | ReLU |
| Dense | | | 7*7*128 | √ | ReLU |
| Transposed Convolution | $4 \times 4$ | $2 \times 2$ | 64 | √ | ReLU |
| Transposed Convolution | $4 \times 4$ | $2 \times 2$ | 1 | × | Tanh |
| *D(x)* | | | | | |
| Convolution | $4 \times 4$ | $2 \times 2$ | 64 | × | Leaky ReLU |
| Convolution | $4 \times 4$ | $2 \times 2$ | 64 | √ | Leaky ReLU |
| Dense* | | | 1024 | √ | Leaky ReLU |
| Dense | | | 1 | × | Sigmoid |
| Optimizer | Adam($\alpha = 10^{-5}, \beta_1 = 0.5$) | | | | |
| Batch size | 100 | | | | |
| Latent dimension | 200 | | | | |
| Epochs | 100 | | | | |
| Leaky ReLU slope | 0.1 | | | | |
| Weight, bias initialization | Isotropic gaussian ($\mu = 0, \sigma = 0.02$), Constant(0) | | | | |

Table 3: MNIST GAN Architecture and hyperparameters

| Operation | Kernel | Strides | Features Maps/Units | BN? | Non Linearity |
|---|---|---|---|---|---|
| *E(x)* | | | | | |
| Convolution | $3 \times 3$ | $1 \times 1$ | 32 | × | Linear |
| Convolution | $3 \times 3$ | $2 \times 2$ | 64 | √ | Leaky ReLU |
| Convolution | $3 \times 3$ | $2 \times 2$ | 128 | √ | Leaky ReLU |
| Dense | | | 200 | × | Linear |
| *G(z)* | | | | | |
| Dense | | | 1024 | √ | ReLU |
| Dense | | | 7*7*128 | √ | ReLU |
| Transposed Convolution | $4 \times 4$ | $2 \times 2$ | 64 | √ | ReLU |
| Transposed Convolution | $4 \times 4$ | $2 \times 2$ | 1 | × | Tanh |
| *D(x)* | | | | | |
| Convolution | $4 \times 4$ | $2 \times 2$ | 64 | × | Leaky ReLU |
| Convolution | $4 \times 4$ | $2 \times 2$ | 64 | √ | Leaky ReLU |
| *D(z)* | | | | | |
| Dense | | | 512 | × | Leaky ReLU |
| *Concatenate D(x) and D(z)* | | | | | |
| *D(x,z)* | | | | | |
| Dense* | | | 1024 | × | Leaky ReLU |
| Dense | | | 1 | × | Sigmoid |
| Optimizer | Adam($\alpha = 10^{-5}, \beta_1 = 0.5$) | | | | |
| Batch size | 100 | | | | |
| Latent dimension | 200 | | | | |
| Epochs | 100 | | | | |
| Leaky ReLU slope | 0.1 | | | | |
| Weight, bias initialization | Isotropic gaussian ($\mu = 0, \sigma = 0.02$), Constant(0) | | | | |

Table 4: MNIST BiGAN Architecture and hyperparameters

## D    KDD99 EXPERIMENT DETAILS

**Preprocessing:** The dataset contains samples of 41 dimensions, where 34 of them are continuous and 7 are categorical. For categorical features, we further used one-hot representation to encode them; we obtained a total of 121 features after this encoding. Then, we applied min-max scaling to derive the final features.
The outputs of the starred layer in the discriminator were used for the *FM* scoring variant.

| Operation | Units | Non Linearity | Dropout |
|---|---|---|---|
| *G(z)* | | | |
| Dense | 64 | ReLU | 0.0 |
| Dense | 128 | ReLU | 0.0 |
| Dense | 121 | Linear | 0.0 |
| *D(x)* | | | |
| Dense | 256 | Leaky ReLU | 0.2 |
| Dense | 128 | Leaky ReLU | 0.2 |
| Dense* | 128 | Leaky ReLU | 0.2 |
| Dense | 1 | Sigmoid | 0.0 |
| Optimizer | \multicolumn{3}{l}{Adam($\alpha = 10^{-5}$, $\beta_1 = 0.5$)} | |
| Batch size | 50 | | |
| Latent dimension | 32 | | |
| Epochs | 50 | | |
| Leaky ReLU slope | 0.1 | | |
| Weight, bias initialization | Xavier Initializer, Constant(0) | | |

Table 5: KDD99 GAN Architecture and hyperparameters

| Operation | Units | Non Linearity | Dropout |
|---|---|---|---|
| *E(x)* | | | |
| Dense | 64 | Leaky ReLU | 0.0 |
| Dense | 32 | Linear | 0.0 |
| *G(z)* | | | |
| Dense | 64 | ReLU | 0.0 |
| Dense | 128 | ReLU | 0.0 |
| Dense | 121 | Linear | 0.0 |
| *D(x)* | | | |
| Dense | 128 | Leaky ReLU | 0.2 |
| *D(z)* | | | |
| Dense | 128 | Leaky ReLU | 0.2 |
| *Concatenate D(x) and D(z)* | | | |
| *D(x,z)* | | | |
| Dense* | 128 | Leaky ReLU | 0.2 |
| Dense | 1 | Linear | 0.0 |
| Optimizer | \multicolumn{3}{l}{Adam($\alpha = 10^{-5}$, $\beta_1 = 0.5$)} | |
| Batch size | 50 | | |
| Latent dimension | 32 | | |
| Epochs | 50 | | |
| Leaky ReLU slope | 0.1 | | |
| Weight, bias initialization | Xavier Initializer, Constant(0) | | |

Table 6: KDD99 BiGAN Architecture and hyperparameters

