# OpenReview forum: "Efficient GAN-Based Anomaly Detection"
_ICLR.cc/2018/Workshop — Reject_

### Official Review · AnonReviewer4 · 2018-03-06
**The result is not very significant.**

**Rating:** 4
**Confidence:** 3

**Review:**

In this paper, the authors proposed using GAN with additional encoders for anomaly detection.
The authors claimed that the proposed method is computationally efficient, and is suitable for anomaly detection in high dimensions.

[Clarity]
Good. The authors described the basic idea clearly.

[Quality, Originality]
GAN with additional encoders is previously studied.
The contribution of this study is showing that such encoder is useful for accelerating the GAN-based anomaly detection.
The idea is simple yet it seems reasonable.
However, technically, it is a direct application of the existing method to anomaly detection application.
I therefore think the originality is low.

[Significance]
The authors claimed that GAN is good at modeling high dimensional distributions.
However, the experiments are conducted only on MNIST and KDD99, which are not very high dimensional.
It is therefore unclear whether the proposed method is really helpful for anomaly detection in high dimensions.
Overall, I think the experimental evaluations failed to support the main claim, and thus it is far from satisfactory.

---

### Official Review · AnonReviewer2 · 2018-03-07
**Review for Efficient GAN-Based Anomaly Detection**

**Rating:** 9
**Confidence:** 5

**Review:**

This work provides a new framework to perform anomaly detection (one-class learning) based on GAN. It used an encoder from the HD space to the latent space that is learned at the same time as the GAN.

Pros:
- Well written, easy to read
- Interesting used of encoder learning with the help of GAN (link to the old-fashion way of using AE  for anomaly detection)

Cons:
-  Author should highlight the difference between their framework and the framework presented in Schlegl 2017 and most notably how to explain such difference in processing time.  (It is not clear enough in the present work that Schlegl 2017 don't used encoder but backpropagated to the latent space which is costly in time)

---

### Official Review · AnonReviewer1 · 2018-03-10

**Rating:** 5
**Confidence:** 3

**Review:**

This paper describes use biGAN for Anomaly detection, followed by AnoGAN. The first step is training a biGAN style GAN with normal data, then define a score function that measures anomalous. The score function is a combination of reconstruction and a loss function for anomaly. Anomaly loss can be cross-entropy or feature map loss. The experiment is performed on MNIST and KDD99.

First concern is, GAN is used to capture training data distribution. In both AnoGAN and this work, the anomaly detection part is only performed at the score function. Which means, the GAN part can be replaced by any encoder-decoder model. The authors may need further explanation or experiment to show why it is GAN but not other model.

Second concern is, the experiment is not convincing. According to experiment table, DAGMM performs really well, and this work is not significantly better than DAGMM. Moreover, there is no explanation of what is full name of DAGMM.

Overall I think this work is below acceptance threshold and there is more space to improve.

---

### Decision · Program_Chairs · 2018-03-20
**ICLR 2018 Workshop Acceptance Decision**

**Decision:**

Reject

**Comment:**

Based on the reviews, this paper has not been accepted for presentation at the ICLR workshop. However, the conversation and updates can continue to appear here on OpenReview.